# First Look into the Use of Fish Scales as a Medium for Multi-Hormone Stress Analyses

**Emily K. C. Kennedy [1],* and David M. Janz [2]**

1   Toxicology Graduate Program, University of Saskatchewan, 44 Campus Drive, Saskatoon, SK S7K 5B3, Canada
2   Western College of Veterinary Medicine and Toxicology Centre, 52 Campus Drive,
    Saskatoon, SK S7N 5B4, Canada; david.janz@usask.ca
*   Correspondence: ekk048@usask.ca; Tel.: +(306)-966-4147

**Abstract:** Recent efforts have provided convincing evidence for the use of fish scale cortisol concentration in the assessment of long-term stress in fishes. However, cortisol alone is not sufficient to fully describe this state of long-term stress. Dehydroepiandrosterone (DHEA) is an androgen with actions that oppose those of cortisol. The means by which DHEA negates the effects of cortisol occurs in part via changes in the metabolism of cortisol to cortisone. The quantitation of cortisol, DHEA and cortisone could therefore provide a more comprehensive assessment of the overall status of physiological stress. As DHEA and cortisone have yet to be quantified within the fish scale, our first objective was to ensure our sample processing protocol for cortisol was applicable to cortisone and DHEA. Following this, we induced a state of long-term stress in goldfish (*Carassius auratus*). Some degree of elevation in all hormones was observed in the stressed fish scales. Additionally, cortisol and cortisone were significantly elevated in the stressed fish serum in comparison to controls while DHEA was undetectable in either group. Overall, these results suggest that fish scales provide an appropriate medium for the assessment of long-term stress in fishes via the quantitation of relevant steroid hormones.

**Keywords:** biomonitoring; conservation physiology; glucocorticoids; physiological stress

## 1. Introduction

The quantification of cortisol in an effort to assess the state of stress in fish has been carried out in myriad media. Serum or plasma is most common, but others include feces, urine, mucus and surrounding water [1–4]. While many of these methods have proven successful, the use of fish scale hormone concentrations in the assessment of stress in fish presents some unique and useful features [5]. Similar in concept to hair or feathers which are already used for hormone quantification purposes, scales have been shown to incorporate cortisol over long periods of time [5–10]. The cortisol concentration of a scale sample is thus an integration of hormones secreted by the hypothalamic–pituitary–inter-renal (HPI) axis over weeks and perhaps even months rather than a single time point as is characteristic of other media [5,6]. Additionally, the cortisol concentration of the scale has been shown to be unaffected by brief increases in stress incurred upon capture, a problem often encountered when sampling blood or mucus [6]. This is also relevant when considering diurnal hormone fluctuations. Unlike blood samples, the time of day at which scale samples are collected is unlikely to have a significant effect on the hormone concentration of the scales [11]. Thus, along with their relative ease of collection, scales appear to provide a useful and convenient means of assessing long-term stress in teleost fishes.

Cortisol is the primary glucocorticoid in fishes and a crucial mediator of the physiological stress response [12,13]. As such, scale cortisol concentration has been shown to increase as a result of external injuries [14], increased water temperature [10], high stocking densities and changes in feeding strategies [9] as well as general long-term stress [5,6],

providing promising evidence for its use as a non-lethal biomarker of long-term stress in both wild and aquacultural fish populations. However, while the quantitation of secreted cortisol provides some indication of the state of stress within an organism, the use of multi-hormone analyses may be capable of uncovering further information [6,15,16]. Dehydroepiandrosterone (DHEA) is an androgen and precursor steroid with actions that oppose those of cortisol in mammals [17–19]. DHEA circulates in the blood in its inactive sulfated form DHEA-S and can later be de-sulfated in order to carry out its function [20,21]. Currently, exploration into the involvement of DHEA in the stress response in fishes is lacking. However, in humans and other vertebrates, high ratios of cortisol to DHEA have been considered indicative of chronic stress and an increased allostatic load [15,16,22]. Although not fully understood, the means by which DHEA negates the effects of cortisol in mammals have been shown to occur in part via changes in cortisol metabolism [23]. This can result from a variety of mechanisms, including increases in the transcription and activity of 11 beta-hydroxysteroid dehydrogenase 2 (11βHSD 2), which converts cortisol to its inactive metabolite cortisone [18,23,24]. During periods of stress, the conversion of cortisol to cortisone via the 11βHSD 2 enzyme could be enacted in order to protect sensitive organs from cortisol surges [23,25]. Reports of increases in rainbow trout 11βHSD 2 equivalent (11rtHSD 2) activity associated with stress in rainbow trout (*Oncorhynchus mykiss*) suggest that an increased cortisone:cortisol ratio could also be an indicator of elevated stress in fishes [25]. Additionally, some studies report cortisol + cortisone concentration to better assess total cortisol secretion [26–28]. Kapoor et al. (2018) demonstrated that circulating cortisol is incorporated into the hair shaft as cortisone by injecting rhesus monkeys with radio-labelled cortisol [29]. Thus, the scale cortisol + cortisone value is likely a better estimate of the total magnitude of the stress response than either hormone alone. The quantification of scale cortisol in addition to scale DHEA and cortisone should therefore provide a more complete picture of the overall state of stress in teleost fish [6].

To our knowledge, DHEA and cortisone have not previously been quantified in fish scales. The first objective of this study was thus to ensure that our sample processing protocol previously used for scale cortisol quantification was applicable to these additional hormones. Following this, we induced a state of long-term stress in goldfish, a commonly used teleost for endocrinological research, in order to analyze changes in scale and circulating cortisol, cortisone and DHEA. Upon completion of the stress protocol, both scale and serum were collected and analyzed for cortisol, cortisone and DHEA concentration as well as DHEA-S in the case of serum. These values were then used to generate multi-hormone values and assess the state of stress in the goldfish. We found that DHEA and cortisone were in fact quantifiable within the fish scale and that long-term stress resulted in elevated scale cortisol and cortisone as well as a significant increase in the scale cortisol:DHEA ratio. Serum cortisol and cortisone concentrations were also significantly elevated, whereas DHEA was undetectable.

## 2. Materials and Methods

### 2.1. Wash Protocol Validation

Our preliminary studies have demonstrated that methanol was effective in removing external cortisol contamination from scales. However, to ensure the efficacy of methanol as a wash solvent for the removal of external cortisol, cortisone and DHEA from goldfish scales, $n = 3$ replicates of a subset of six scale samples were washed one to six times, and the cortisol, cortisone and DHEA content of all six wash solutions as well as their matching scale samples was measured. Scale samples of 200 mg were placed into a 5 mL plastic tube with 4 mL of methanol and vortexed for two and a half minutes. These masses were deemed appropriate by analyzing four sub-samples ranging from 25 to 100 mg in order to determine how much dry scale mass was required to create a sample extract sufficiently concentrated to fall within the linear section of the standard curve created with each ELISA. Between each wash, methanol was decanted, scales were blotted dry, and any visible debris (skin, etc.) was removed with forceps. Wash tubes were also rinsed between each wash,

and a fresh aliquot of methanol was used for each successive wash. After the respective number of washes, the final wash solution for each scale sample was collected into a glass culture tube, and scales were placed in a filter-paper-lined Petri dish with the lid off-set for air flow and allowed to dry on the bench top for 24 h. Collected wash solutions were dried at 38 °C under a gentle stream of nitrogen gas. These tubes were then rinsed four times (1 mL, 0.4 mL, 0.2 mL and 0.15 mL) alongside the matching scale sample extracts as described in the extraction process outlined in following sections.

### 2.2. Stressor Exposure

Goldfish were in the regressed stage (April) and approximately 5 cm in length. A total of 56 goldfish were subdivided into two groups of *n* = 28, one of which was subjected once daily to a stressor for 14 days and one of which served as a control. The repetitive application of acute stressors can be used to bring about a state of chronic stress [30]. As such, one of three different stressors previously shown to generate an acute stress response in fish by Aerts et al. (2015) was randomly applied to the stressed group: (1) holding above water for two min, (2) chasing for 10 min with a net or (3) holding in a bucket with an insufficient amount of water for five min [5]. The stressor was also applied at a randomly assigned time of day (9 a.m., 12 p.m. or 3 p.m.) in an effort to maintain the unpredictability of the stressor.

### 2.3. Sample Collection

Fish were anesthetized using buffered MS-222 (100 mg/L). A sample of blood was then collected into hematocrit tubes via caudal severance and dispensed into Eppendorf tubes. Blood was left to clot for 3 h on ice and then centrifuged to allow the collection of serum, which was stored at −20 °C until further analyses. Prior to scale collection, fish were euthanized via cervical severance and wiped down to remove excess mucus. Scales were then collected by scraping the length of the body towards the tail with a metal spatula and stored at −20 °C until further analyses.

### 2.4. Hormone Extraction and Quantitation

Based on preliminary work conducted in goldfish scales, the analysis of each hormone concentration required 50 mg of dry powdered scale. Due the small size of the fish used in this experiment, the scales of two fish were necessary to meet this requirement. Thus, to ensure the proper analysis of cortisol-to-DHEA hormone ratios, the powdered scale from four goldfish was pooled and divided into two 50 mg subsamples: one for cortisol analysis and one for DHEA analysis. Any remaining scale mass was used for cortisone analysis.

To remove external contaminants and ensure accuracy of internal scale cortisol concentrations, all scale samples were briefly washed three times with methanol as described above. Washed and dried scales were then ground to a fine powder using a Retsch ball mixer mill. Samples were ground in a 10 mL grinding jar with a 12 mm stainless-steel grinding ball for 0.045 s per mg of scale at 30 Hz. Subsamples of 50 mg were then combined with 1 mL of methanol and vortexed briefly for 15 s. Tubes were then placed in a rotator and left for 18 h to extract at room temperature. Extracted samples were centrifuged for 15 min at 4500 rpm and 20 °C, and extracts were collected into 5 mL borosilicate glass tubes and dried at 38 °C under a gentle stream of nitrogen gas. A second 1 mL aliquot of methanol was added back to the powdered samples and vortexed for 40 s, then centrifuged, collected and evaporated as above. These steps were repeated twice for a total of three collections. To concentrate extracted cortisol at the bottom of each tube, the sides were rinsed four times with decreasing volumes of methanol (1 mL, 0.4 mL, 0.2 mL, 0.15 mL). Between each rinse, extracts were dried at 38 °C under nitrogen gas. Final extracts were then reconstituted in 200 μL of buffer supplied by their respective EIA kits: Cortisol EIA kit (Oxford Biomedical, Rochester Hills, MI, USA), Salivary DHEA Enzyme Immunoassay Kit (Salimetrics®, Carlsbad, CA, USA) or DetectX® Cortisone Enzyme Immunoassay Kit (Arbor-Assays®, Ann Arbor, MI, USA). Serum collected from individual fish was pooled in the

same manner as the scale samples in order to create matching pairs. Serum samples were prepared for analysis using the protocols outlined in the appropriate ELISA kit. In addition to serum DHEA, we also attempted to quantify serum DHEA-S using the DHEA-S Enzyme Immunoassay Kit (Arbor-Assays®). Finally, all samples were run in triplicate following the kit protocols in a Molecular Devices Spectra Max 190 microplate spectrophotometer.

Extracts from multiple samples were pooled for intra-assay variation (*n* = 5) and inter-assay variation (*n* = 10), determined as the percent coefficient of variation (%CV, SD/mean). Intra- and inter-assay variation for scale cortisol concentration was 3.9% and 11.9%, respectively, and 6.7% and 8.8% for serum cortisol, respectively. Intra- and inter-assay variation for scale cortisone concentration was 3.7% and 11.8%, respectively, and 2.7% and 8.2% for serum cortisone, respectively. For scale DHEA concentration, intra- and inter-assay variation was 6.5% and 10.1%, respectively. Parallelism between extracted samples and the kit standard curve was determined using a serial dilution of the pooled extract run in triplicate. Parallelism was observed between all standard curves and serially diluted extracts generated from both scale and serum samples. This validation excluded serum DHEA and DHEA-S as they were undetectable. Limits of detection (LODs) for cortisol, cortisone and DHEA kits were 0.00510 ng/mL, 0.0285 ng/mL and 0.00127 ng/mL, respectively. Any extracts with a hormone concentration below the limit of detection were assigned the limit of detection concentration. While the LOD for cortisone was higher than LODs for cortisol and DHEA, no scale hormone concentrations in either the control or stressed group were below detection for any hormone; this only occurred in the wash protocol validation.

### 2.5. Statistical Analyses

Prior to any statistical testing all data were tested for normality and homoscedasticity using the Shapiro–Wilk test and Bartlett's test, respectively, as well as a visual inspection of the residuals. If parametric, comparisons of wash solution and scale extract hormone concentrations in the wash dynamics study were assessed using one-way ANOVA, and multiple comparisons were assessed using a Tukey test. If non-parametric, a Kruskal–Wallis test was employed followed by a Dunn's test for multiple comparisons. In the case of scale and blood hormone concentrations, comparisons between control and stressed groups were performed using an unpaired T-test if data sets were parametric. If non-parametric, a Mann–Whitney test was employed. Differences between groups were deemed significant at $p < 0.05$.

## 3. Results

### 3.1. Wash Protocol Validation

The cortisol concentration of the first wash solution (10.8 ng/mL) was significantly greater than all subsequent wash solutions ($p < 0.05$); however, the cortisol concentration among washes two to six did not differ significantly (Figure 1). Cortisone and DHEA followed the same pattern, with concentrations in the first wash solutions (5.00 ng/mL and 0.0454 ng/mL, respectively) being significantly greater when compared to all other wash solutions (Figure 1; $p < 0.05$). In the case of cortisol and DHEA, the hormone concentrations among scale samples one to six were not significantly different (Figure 1). In the case of cortisone, the hormone concentration of scale sample one was significantly greater than scale samples three to six ($p < 0.05$); however, cortisone concentration among samples two to six did not differ significantly (Figure 1).

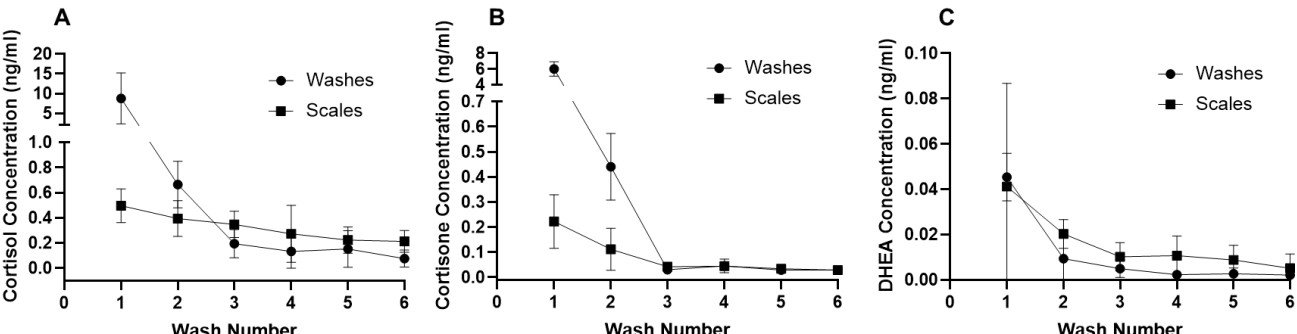

**Figure 1.** Validation of scale washing procedure: (**A**) cortisol, (**B**) cortisone and (**C**) DHEA concentrations in goldfish scale sample extracts washed 1–6 times are presented alongside their matching wash solution extracts. Error bars represent standard deviations; see text for further description of statistical analyses (*n* = 3).

*3.2. Scale Hormone Concentrations*

Cortisol, cortisone and DHEA concentrations were all somewhat elevated in scales collected from stressed goldfish when compared to control goldfish scales (Figure 2). While none of these elevations were statistically significant, the comparison of scale cortisol and cortisone concentrations between control and stressed goldfish produced notable *p*-values of 0.052 and 0.071, respectively. However, the cortisol:DHEA ratio was significantly elevated in the stressed group in comparison to control (*p* < 0.05, Figure 3).

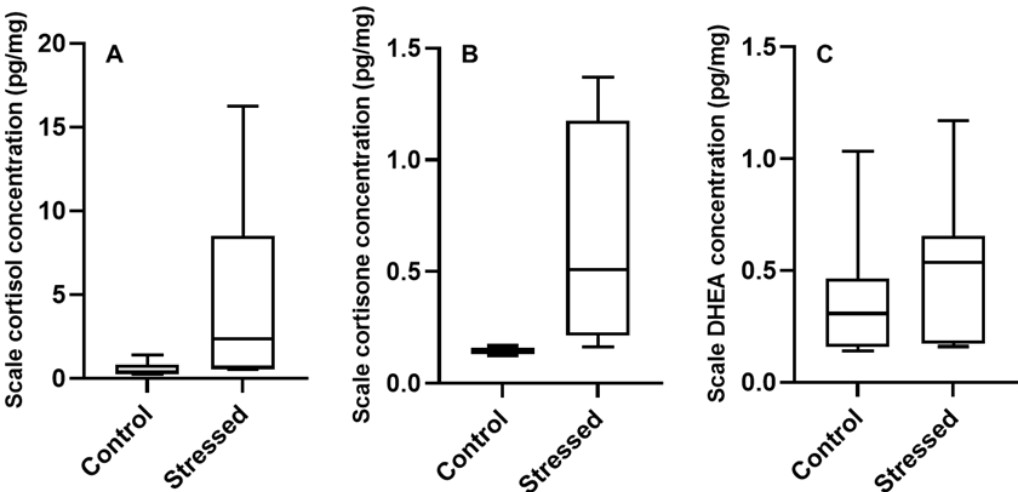

**Figure 2.** Scale hormone concentrations in control and stressed goldfish: (**A**) cortisol, (**B**) cortisone and (**C**) DHEA concentrations presented as the median (middle line), surrounded by the 95% confidence interval (rectangle) with whiskers extending to the full range of the data. Differences between control and stressed goldfish scale hormone concentrations were not significant (*p* > 0.05; *n* = 8–14).

*3.3. Serum Hormone Concentrations*

Serum hormone concentrations are presented in Figure 4. Cortisol and cortisone concentrations were significantly elevated in the stressed goldfish serum (*p* < 0.05). DHEA and DHEA-S were undetectable in serum collected from either the control or stressed group. While the cortisol + cortisone value was significantly elevated in stressed goldfish serum (*p* < 0.05), the cortisone:cortisol ratio was not significantly different between the two groups (Figure 5).

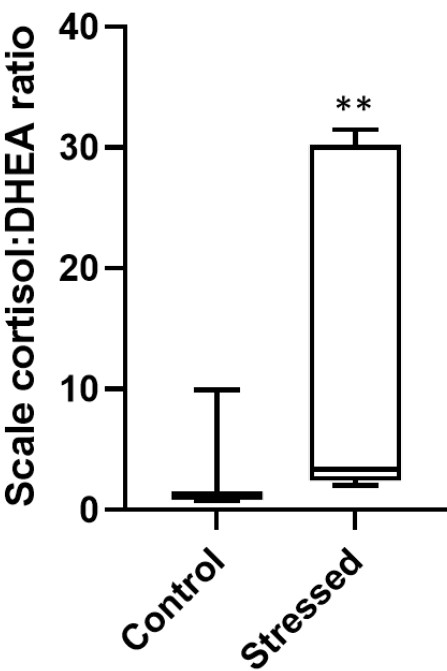

**Figure 3.** Scale cortisol:DHEA ratio in control and stressed goldfish presented as the median (middle line), surrounded by the 95% confidence interval (rectangle) with whiskers extending to the full range of the data. Asterisks indicate significant difference from control ($p < 0.01$; $n = 14$).

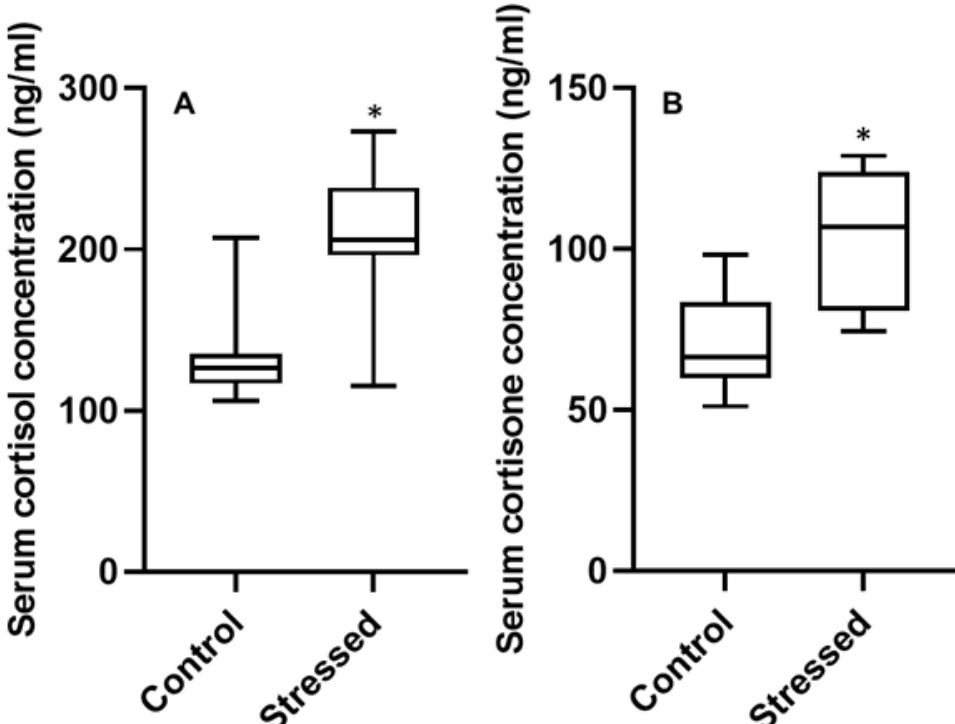

**Figure 4.** Serum hormone concentrations in control and stressed goldfish: (**A**) cortisol and (**B**) cortisone concentration presented as the median (middle line), surrounded by the 95% confidence interval (rectangle) with whiskers extending to the full range of the data. Asterisks denote significant differences from control ($p < 0.05$; $n = 14$).

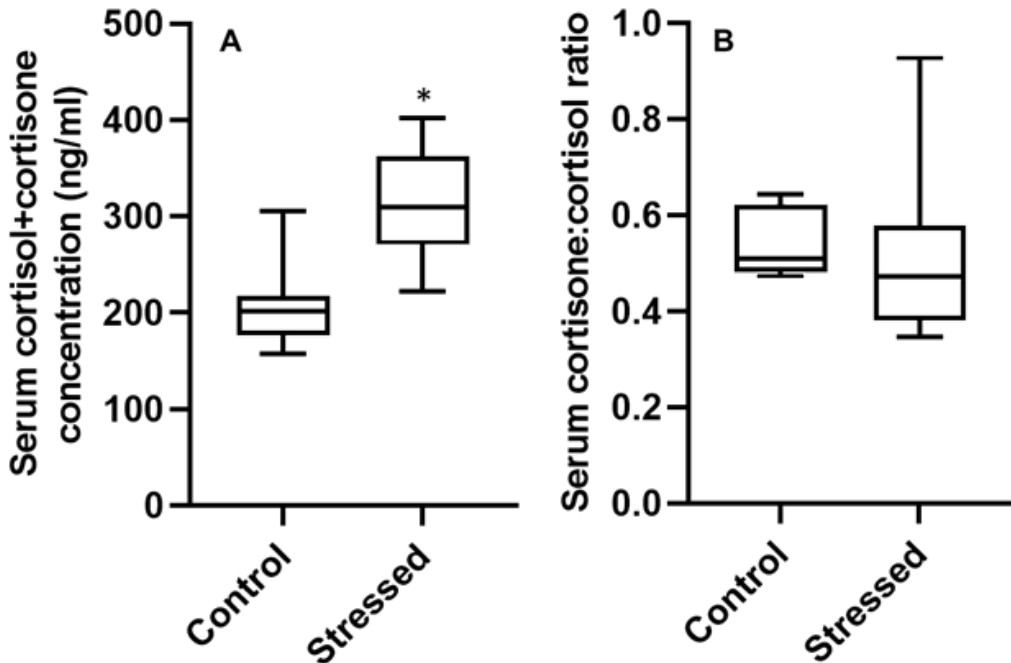

**Figure 5.** Multi-hormone values calculated from goldfish serum hormone concentrations: (**A**) serum cortisol + cortisone concentration and (**B**) serum cortisone:cortisol ratio in control and stressed goldfish presented as the median (middle line), surrounded by the 95% confidence interval (rectangle) with whiskers extending to the full range of the data. Asterisk indicates a significant difference from control ($p < 0.05$; $n = 14$).

## 4. Discussion

To our knowledge, this is the first study quantifying cortisone and DHEA in fish scales. As our results demonstrate, these hormones were sufficiently removed from the scale surface by washing with methanol and were capable of being extracted from powdered scale in the same manner as cortisol. In all cases, two washes appeared to be sufficient for the removal of external contaminants. However, in some cases scale hormone concentrations fell below detection limits after three washes. This indicates that there is potential for leaching of internal hormone content when more than three washes are employed. We thus recommend a minimum of two and a limit of three washes with methanol per scale sample. With these methods, we are now able to generate multi-hormone data useful in the assessment of long-term stress in fish, as was also accomplished during this study.

In order for scale hormone concentrations to serve as a means of assessing stress in teleost fishes, hormones must partition proportionally from blood to scale; however, the relationship between circulating and scale hormone concentrations has yet to be fully elucidated. While concurrent changes in both scale and serum collected from the same organism could aid in confirming their proportionality, due to the rapid and frequent changes in circulating hormone concentrations, this may not always be the case [8]. Neither DHEA nor DHEA-S were detectable in any of the goldfish serum samples, yet scale samples collected from both stressed and unstressed fish contained measurable concentrations of DHEA. As DHEA-S is known to circulate in relatively low concentrations in many fish species, this could aid in confirming that fish scales are gradually accumulating DHEA and likely other steroid hormones over time [31]. By contrast, the significant increases in stressed goldfish serum cortisol and cortisone were not wholly reflected in scale samples; nevertheless, notable elevations in stressed fish scale cortisol and cortisone were observed. Results presented by other groups suggest that this could be due to a lag in the transfer of hormone from blood to scale [5,6]. However, there are other phenomenon capable of disrupting blood-scale proportionalities. The 11-βHSD 2 enzyme responsible for the conversion of cortisol to cortisone can be found within fish skin [32]. Similar to the

peripheral hypothalamic pituitary adrenal axis present within hair follicles in mammals, this raises concerns regarding preferential deposition of locally generated metabolites within the scale as this could interfere with blood-scale proportionalities [6,32,33]. Additionally, as is the case in other cumulative media such as hair and feathers, the partitioning of steroid hormones from blood to scale likely occurs via passive diffusion [29,34]. Slight variances in the chemical properties of different steroid hormones could therefore increase their degree and depth of incorporation into the scale, as has been reported in hair [15]. Likewise, the rate of hormone clearance from both media must also be considered. Scale composition is not wholly static as scales participate in select physiological processes such as calcium homeostasis [35]. As such, many factors could influence the concentration of hormone residing in the scale at the time of sampling, including a redistribution of hormones from scale to blood. In depth mechanistic studies exploring scale steroid hormone deposition and elimination are essential in answering these questions.

Unlike individual scale hormone concentrations, the scale cortisol:DHEA ratio was significantly elevated in stressed fish, suggesting a potentially more robust marker of chronic stress [15,16,22]. As cortisol and DHEA have been shown to counteract one another in mammals, it is also likely that their net activity, represented by the cortisol:DHEA ratio, better describes the state of stress experienced by an organism than either hormone alone. Other multi-hormone analyses potentially useful in the assessment of long-term stress include the cortisol + cortisone concentration and the cortisone:cortisol ratio. The conversion of cortisol to inactive cortisone is thought to be enacted during stressful periods in order to protect sensitive organs such as the gonads [25,36,37]. Unfortunately, these two values were difficult to calculate using our scale hormone data as we lacked sufficient powdered scale to measure matching cortisol and cortisone concentrations as were measured for cortisol and DHEA. Our sample size for scale cortisone concentration was also lesser than those of the other two hormones. Still, the average cortisol + cortisone concentration was greater in the stressed group (5.96 pg/mg) when compared to the control group (0.713 pg/mg). By contrast, the average scale cortisone:cortisol ratio was lesser in the stressed group (0.123) than in the control group (0.256). As we were able to collect a sufficient volume of serum to measure matching cortisol and cortisone concentrations, multi-hormone serum comparisons were more easily generated. Similar to the scale cortisone:cortisol ratio, the serum cortisone:cortisol ratio was slightly lesser in stressed goldfish; however, this difference was not statistically significant. As the conversion of cortisol to cortisone has been shown to increase with increased stress, this decrease in scale and serum cortisone:cortisol ratios was unexpected; however, the concurrent decrease in both media could add evidence that these hormones are depositing within the scale proportional to circulating concentrations. The conversion of cortisol to cortisone is also particularly relevant to reproduction as previously mentioned [25,38]. As our fish were in the regressed stage, this value may be less relevant to the state of stress in goldfish used in the present study.

Similar to the scale cortisol + cortisone concentration, serum cortisol + cortisone was significantly elevated in the stressed fish. Although cortisone is an inactive compound that no longer participates in the stress response, the relationship between cortisol and cortisone via the 11-βHSD 2 enzyme maintains cortisone's relevance in the assessment of stress. Thus, the combined cortisol + cortisone value likely provides a better estimation of the total glucocorticoid release and HPI axis activity than cortisol alone [29,39]. Alongside the cortisol:DHEA ratio, this amplification of the stress response created by the use of hormone ratios and other multi-hormone values is of potential benefit to this area of research [15]. Ultimately, our goal in the development of these non-lethal measures of long-term stress is to conserve and protect fish populations. Stress responsiveness varies greatly both inter- and intra-specifically, making statistical comparisons between control and stressed groups difficult to analyze [30,40]. Power analyses suggest doubling or tripling our sample size would be necessary to conduct more robust statistical analyses. However, by using hormone ratios and increasing our ability to detect meaningful differences between stressed

and unstressed organisms, we may be able to reduce the required sample size, decreasing negative impacts on future study populations.

## 5. Conclusions

This study validated a laboratory technique to quantify cortisol, DHEA and cortisone in scales collected from goldfish, then applied the measurement of these steroid hormones to the assessment of long-term stress in fishes. Although there are many knowledge gaps left to be filled, the results offer evidence of the practicality of using scale hormone concentrations in the assessment of long-term stress in teleost fish. While the sample sizes used in this experiment were not sufficient to provide concrete conclusions, our results suggest that multi-hormone analyses could be more revealing of the state of stress in fish than single-hormone values. Altogether, the use of scale multi-hormone values in the monitoring of stress in teleost fish has potential as an important tool for the conservation of teleost fishes.

**Author Contributions:** Conceptualization, E.K.C.K. and D.M.J.; methodology, E.K.C.K. and D.M.J.; validation, E.K.C.K.; data curation, E.K.C.K.; writing—original draft preparation, E.K.C.K.; writing—review and editing, D.M.J.; supervision, D.M.J.; funding acquisition, D.M.J. All authors have read and agreed to the published version of the manuscript.

**Funding:** This research was funded by a Natural Sciences and Engineering Research Council of Canada (NSERC) Discovery Grant to D.M.J. (RGPIN-2016-05131).

**Institutional Review Board Statement:** This study was conducted under the guidelines of the Canadian Council on Animal Care and approved by the Animal Research Ethics Board of the University of Saskatchewan (approval code 2020-0118, approval date 8 December 2021).

**Conflicts of Interest:** The authors declare no conflict of interest.

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
