# Peer review of "First Look into the Use of Fish Scales as a Medium for Multi-Hormone Stress Analyses"

_fishes, doi:10.3390/fishes7040145_

Round 1
Reviewer 1 Report
This is a very interesting paper on the subject of non lethal sampling and provides some methodological insights on the measurements of stress hormone levels in fish scales in a two step process first the analytical validation of the detection method and second the induction of stress response hormones in an experiment. The experiments are well described, however some more insights on the QA/QC of the analysis would be useful for improving the confidence in the methods proposed. There are differences in the detection limits of the methods used in more than one order of magnitude (L 164), that fact could be discussed in the paper since is crucial to the applicability of the proposed method. On the other hand, the data manipulation should be more transparent in terms of how many ND values were used for the statistical analysis in each experiment (lines 164-165), and what happened if these values are not considered in the statistics. The hormones response variability of the stress experiments seems to be high, but probably as a result of the stress methods used over time. Did the authors check if the stress method results in different outcomes?: I suggest to better develop this section and indicates in a more explicit way how many values were below detection limits.The paper will benefit from these clarifications since the methods proposed may have a great utility in the field of monitoring fish health not only in lab settings but in the field as well. The innovation of detecting hormones in scales is greatly appreciated, but we need some more insights on the analytical methods used and its validation for accomplishing the authors objectives. The issue of samples sizes also merits a further discussion within the paper. Given the results obtained, what should be a valid sampling size? what should be the accepted variability to conclude differences between control and stressed fish?. That questions are still opens but the readers would like to see responded in the paper's text.
Reviewer 2 Report
Comments-Fishes-1741456
Manuscript fishes-1741456 is a study evaluating the quantity of stress-related hormones (cortisol, DHEA, and cortisone) in scales collected from goldfish. Hormone levels were then compared to their levels in serum samples. The present study offers that multi-hormone analyses might more reliable for describing the state of stress in fish than single hormone analysis, and this non-lethal technique could be used a potential tool for detecting a long-term stress in fish. However, in my perspective the current form of ms is not suitable for publication yes. Improvement should be done by addressing a number critical issues found in the text. In addition, text will be benefit after rewriting some parts for English use, expression, etc.
Abstract
- The use of notation F and E for cortisol and cortisone respectively can be omitted as the authors did not use them in the body of ms. It is just fine to mention them directly without replacing with notation.
Lines 15-7 - rewrite this sentence!
22 - should be “the quantitation of cortisol and cortisone hormones”
Introduction
Line 64 - include the Latin name for rainbow trout
66 - this sentence is not complete and deserves to be expanded!
76-8 - Justification for using goldfish as a model in this study deserves to be included!
77 - the Latin name for goldfish has been mentioned previously, thus It can be omitted!
Materials and methods
89-91 - modify this sentence!
108 - “…., one of which…”
112 - was randomly
119-20 - was the blood left to clot for some time between collection and centrifugation?
146-8 were the kits validated for fish/goldfish?
Results
180 - Please check cortisol units here, I think It should be “nmol/L”. See also line 211
193 - What do the Bars mean? SD or SE? State it
Discussion
- The text must be rewritten for correcting the grammatical errors, for example: lines 231, 232, and 236 (should be “were”), line 233 (appeared), etc.
240-3 - modify this sentence!

Reviewer 3 Report
This works deals with new stress markers in fish. It has two interesting points since introduces some hormones as new potential indicators, and use a non-invasive technique to collect samples. Undoubtedly both issues are innovative and need a more exahustive study.
Therefore this work looks like a "introduction" to new methodolgies to assess chronic stress in fish, since the results are not totally concluding. I hope authors continue researching on this topic to reach relevant results.
Besides the comments I have marked in the atached PDF, I suggest authors to review and include other references on F, E and DHEA in other animals. Maybe those could can help them to justify the lack of changes in some of the trials.

Round 2
Reviewer 1 Report
The authors responded my queries in a satisfactory way, I am satisfied with the paper and I think the paper may be recommended for publication now. The authors did a good job improving the paper and also setting the limitations of the proposed approach.
Author Response
Thank you for your comments and suggestions.
Reviewer 3 Report
The authors have been replied to all my concerns and I think now it is suitable for publication.
Author Response

(The authors gave the same response as above.)
